# Nicotine alters cellular activity and mRNA expression of patterns of Astrocytes

Leslie Sewell[1], James J. Cray[1,2]*

1 Department of Biomedical Education and Anatomy, College of Medicine, The Ohio State University, Columbus, Ohio, United States of America, 2 Divisions of Biosciences and Orthodontics, College of Dentistry, The Ohio State University, Columbus, Ohio, United States of America

* James.Cray@osumc.edu

## Abstract

Nicotine exposure during neural development presents a significant public health concern. Nicotine, the primary addictive component of tobacco, influences the central nervous system by interacting with various cell types, including the glial cell termed astrocytes. Astrocytes are cells that are critical for supporting neurons, regulating neurotransmitter balance, and managing neuroinflammation. This current study explored nicotine's effects on astrocytes, examining cellular activity and gene expression within an acute exposure period. Murine C8D1A astrocytic (garnered as a cell line from postnatal day 8 tissue) cells were treated with nicotine (0–500 ng/mL) in vitro, with assays measuring cell viability and apoptosis at 12, 18, 24, and 48 hours to establish a critical concentration gradient for nicotine. Nicotine exposure increased astrocyte viability at later time points (24 and 48 hours), while apoptosis rose initially but declined over time allowing for the establishment of pharmacologically and clinically relevant nicotine concentrations of 25,50 and 100ng/ml for subsequent experiments. Real-time quantitative PCR revealed that nicotine influenced inflammatory signaling, with pro-inflammatory (A1) markers (IL-6, IFNγ, TNFα) increasing in a dose- and time-dependent manner, while anti-inflammatory (A2) markers (ARG1, IL-10, TGFβ) displayed a more complex pattern after nicotine exposures to astrocytes. These results suggest that nicotine disrupts astrocyte function and inflammatory balance, which may contribute to neurodevelopmental disruptions and heightened neuroinflammatory risks in adults. Further research is needed to investigate the prolonged impact of nicotine on brain health, addiction, and associated neurological conditions.

## Introduction

Use of nicotine containing products and affects in health continues to be a significant public health concern [1–3]. Nicotine is the primary psychoactive compound included in cigarettes, nicotine replacement therapies, and emerging electronic cigarette

**Data availability statement:** All relevant data are within the paper and its Supporting Information files.

**Funding:** The Ohio State University College of Medicine (JC).

**Competing interests:** The authors have declared that no competing interests exist.

delivery systems or "vape" technologies [2–10]. Despite societal endeavors to reduce smoking, nicotine addiction continues to be a major public health crisis.

Although nicotine is infrequently abused in its isolated form, it is predominantly consumed as a constituent of tobacco, most commonly via inhalation of smoke from conventional or electronic cigarettes [10,11]. Tobacco smoke comprises a complex mixture of hundreds of chemical compounds, many of which may potentiate the psychoactive properties of nicotine. Nonetheless, nicotine remains the principal agent underlying tobacco dependence, primarily through its capacity to reinforce drug-seeking and drug-taking behaviors. A key factor contributing to this dependence is nicotine's relatively short plasma half-life of approximately two hours, indicating that half of the administered dose is metabolized and cleared from the body within that timeframe [10]. This rapid pharmacokinetic profile results in a transient duration of action, often compelling individuals to engage in frequent re-administration to sustain its psychoactive effects. Chronic exposure to nicotine induces neuroadaptive changes within the central nervous system, involving alterations across cellular, tissue, and metabolic domains, thereby reinforcing its addictive potential [9,12,13].

One such cell that may be targeted by nicotine is the astrocyte. Astrocytes, resident glial cells of the central nervous system, play crucial roles in supporting neurons, facilitating synapse formation, maintaining the integrity of the blood-brain barrier, regulating neurotransmitter levels, contributing to energy metabolism, and influencing the development of neurological diseases [11–18]. Further astrocytes exert many integral functions during gray and white matter development. After the initial production of neurons, astrocytes can operate as guides for neuronal migration. Behaving as neuronal precursors, radial glia affords a scaffold for neuronal positioning in addition to offering an area in which neurons can travel [19].

Astrocytes undergo diverse morphological, structural, metabolic, and molecular signaling modifications in response to insults including disease and exposures like nicotine [16,20–22]. These adaptations have notable consequences for the typical functioning of astrocytes. These modifications can impact synaptic communication with neurons [20,22–25]. While reward signaling associated with nicotine exposure are intricately associated with the activities of neuronal circuits [21,26], it is reasonable to suggest that astrocytes can exert substantial influence on these neuronal circuitries.

Given the limited data on the specific effects of nicotine on astrocytes, we conducted targeted in vitro studies to determine whether astrocytes are responsive to nicotine and whether nicotine exposure alters their cell cycle in a dose- and time-dependent manner. Based on reported plasma nicotine concentrations following the consumption of a single cigarette (5–30 ng/mL; [1,27–29]), we selected a range of nicotine concentrations (0, 10, 25, 50, 100, 250, and 500 ng/mL) to capture a spectrum of cellular responses in initial screening experiments to establish a concentration gradient of effects. Subsequent analyses focused on concentrations (0, 25, 50, and 100 ng/mL) that elicited measurable cellular activity. We then assessed astrocytic mRNA expression profiles to test the hypothesis that nicotine exposure shifts the balance between neurotoxic/pro-inflammatory and neuroprotective/pro-reparative

markers. Our hypothesis what that nicotine would act in a dose dependent manner to drive greater markers of neuroinflammation over our acute exposure time course.

## Materials and methods

### Cells and culture conditions

Murine astrocytic C8D1A cells [30] were obtained from the American Type Culture Collection (ATCC, USA) and cultured in Dulbecco's modified Eagle's medium (DMEM) supplemented with 10% fetal bovine serum (FBS) and 1% streptomycin/penicillin (Sigma, USA). The cells were maintained at a temperature of 37°C with 5% $CO^2$ with media changes twice per week until 95% confluence was reached. At the time of confluence, cells were seeded at an optimal cell density of 2,500 cells per well for cell proliferation and apoptosis assays. Optimal cell density of 2500 cells per well was determined through preliminary experimentation (Supplementary Fig 1).

### Functional assays

Cell viability was assessed using the CellTiter 96® AQueous One Solution Cell Proliferation Assay (MTS, Promega), which quantifies the conversion of MTS tetrazolium to Formazan—a process directly proportional to the number of viable cells and often prescribed as a measure of cellular proliferation. C8D1A astrocyte cells were seeded in 96-well plates at a density of 2,500 cells per well and treated with nicotine (Sigma Aldrich, St. Louis, MO, USA, N3876) at concentrations ranging from 0 to 500 ng/mL, suspended in standard culture media. Assays were conducted at 12-, 18-, 24-, and 48-hour post-treatment. To evaluate apoptosis under the same treatment conditions, the Apo-ONE® Homogeneous Caspase-3/7 Assay (Promega) was used. At each time point, 100 µL of Caspase-3/7 substrate/buffer solution (1:100 dilution) was added per well. Plates were shaken for 30 seconds at 300 rpm and incubated at room temperature for 1 hour. Fluorescence was then measured using a 96-well plate reader (BioTek) with excitation at 485 nm and emission at 530 nm. All experiments were performed in triplicate. These experiments were utilized to establish the appropriate concentration gradient of effect while maintaining focus on clinically relevant circulating nicotine values.

### RNA isolation

Upon completion of functional assays, RNA studies were devised based on the stablished nicotine concentration gradient. Briefly cells were seeded at a density of 100,000 cells per well in 6 well culture plates and treated for 24-, 48-, and 72-hours with standard culture media supplemented with 0ng/ml, 25ng/ml, 50ng/ml, and 100ng/ml concentration of nicotine (Sigma Aldrich, St. Louis, MO, USA, N3876). RNA was isolated using the OMEGA BioTek E.Z.N.A. Total RNA Kit (Omega BioTek, Norcross, GA) according to manufacturer's protocol. Quality and quantity of RNA was assessed using a Synergy Hi Microplate reader and a Take3 Microvolume Plate (BioTek), with purity being assessed as 260/280 values >2.0. Complimentary DNA synthesis was performed using Quanta qScript cDNA Synthesis reagents following manufacturer's protocol (Quanta Biosciences, Beverly, MA).

### real time quantitative polymerase chain reaction for Astrocyte polarization markers

To quantify the expression levels of targets associated with astrocyte reactivity in response to stress or injury (often termed "polarization"), we conducted quantitative PCR (qPCR) on the complementary DNA (cDNA). For this purpose, we utilized the Applied Biosystems TaqMan Gene Expression Master Mix along with targeted TaqMan gene expression assays. Specifically, we focused on specific astrocyte targets defined as having high expression under neurotoxic or neuroprotective conditions (Table 1). To ensure reliable data interpretation, we employed the ΔCT method for data normalization. In this approach, we used 18S (Mm03928990_g1) ribosomal RNA expression as the reference target. Quantitative data were compared for gene expression changes due to treatment with nicotine by ΔΔCT methodology. Previously

**Table 1. Pro and Anti-Inflammatory Targets with Astrocyte Reactivity.** In the table below are targets that are associated with anti- and pro-inflammatory states after injury or insult.

| Inflammatory Targets | |
| --- | --- |
| **A1** | **A2** |
| IFNγ (Mm01188134_m1) | **ARG1** (Mm00475988_m1) |
| IL-6 (Mm00446190_m1) | **IL-10** (Mm01288386_m1) |
| NOS2 (Mm00440502_m1) | **TGFβ-1** (Mm01178820_m1) |
| TNF (Mm00443258_m1) | **VEGF** (Mm00437306_m1) |

published statistical analysis methodology was used to determine differences for gene expression after nicotine related to the target of interest [31,32]. Differences were considered significant if $p \leq 0.05$.

## Statistical analysis

Quantitative data was compared to control for gene expression after pharmacological exposures for targets of interest. We used statistical analyses for qrt-PCR data utilizing ΔΔCT methodology to assess differences in gene expression. Differences were considered significant if $p \leq .05$. Raw data is provided in supplemental file (S1 Data).

## Results

### Functional assays

C8D1A cell viability assessment following nicotine exposure was assessed using the MTS assay at 12, 18, 24, and 48 hours. Early time points (12 and 18 hours) revealed a modest reduction in cell viability across most nicotine concentrations, with a slight rebound observed at 100 ng/mL at 12 hours and a similar trend at 18 hours following 50 ng/mL treatment. These findings suggest an initial sensitivity of astrocytes to nicotine. By 24 and 48 hours, nicotine exposure exhibited a stimulatory effect on cell proliferation across all tested concentrations. However, this proliferative response plateaued at concentrations above 50 ng/mL, indicating that increasing nicotine concentration beyond 50 ng/mL did not result in proportionally greater increases in cell proliferation—the beneficial effect of nicotine started to level off, or even decrease, despite higher doses (**Fig 1**).

Cell proliferation as indicated by MTS Assay at 12-, 18-, 24- and 48-hour time points. Note the increase by time and relative consistency by nicotine concentration. This suggests a potential effect for nicotine where excessive amounts result in diminishing returns or no greater effects based on MTS cellular assay.

After Caspase 3/7 assay administration, apoptosis levels were measured for nicotine treatment at 12-, 18-, 24- and 48-hour effects. The relationships suggested an initial increase in apoptosis for all concentrations pf nicotine with return to homeostatic levels (similar to control) at 18 hours. At 24 hours little apoptosis was observed for control conditions with noted apoptosis activity in the 10–100 ng/ml nicotine treated concentrations. At 48 hours of note was increase apoptosis activity evidenced with the 25 mg/ml concentrations (**Fig 2**).

Apoptosis of C8D1A Cells as indicated by positive Caspase-3/7 Assay at 12-, 18-, 24- and 48-hour time points. Note the greatest amount of apoptosis occurs rapidly by the 12 hour time point with decreased evidence thereafter. Note similar relationship for nicotine concentration to that observed for MTS assay.

### Nicotine exposure's effect on A1 polarization

To determine the effect of nicotine on astrocyte polarization, C8D1A cells were treated in vitro with 25 ng/ml, 50 ng/ml, and 100 ng/ml nicotine for 24-, 48-, and 72- hours based on derived values for MTS and Caspase experimentation. Assessment of A1 markers at 24 hours indicated a slight stepwise increase in expression were seen with IL-6 ($p = 0.264$, 0.278,

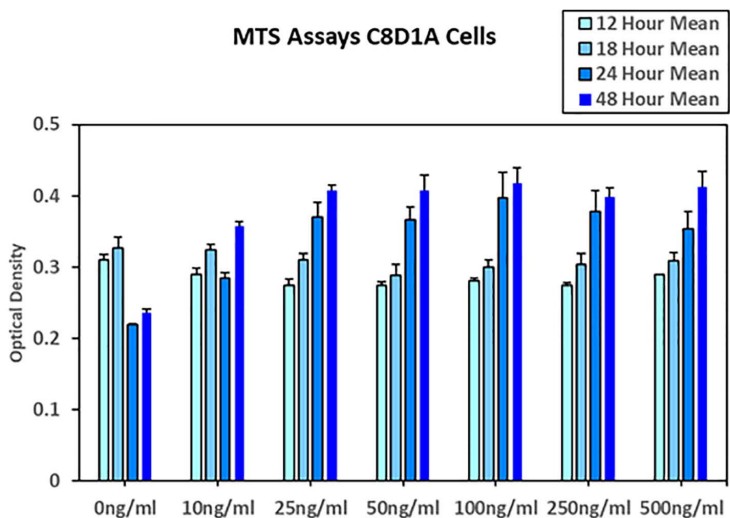

**Fig 1. Cell viability and nicotine responsiveness.**

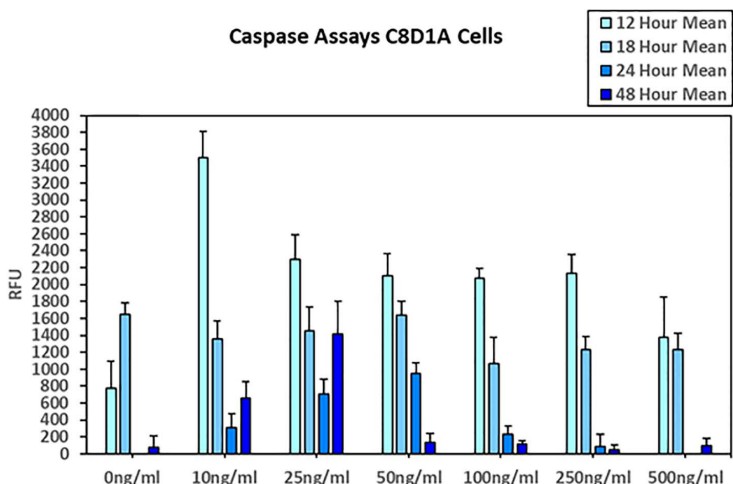

**Fig 2. Apoptosis and Nicotine Responsiveness.**

0.003), IFNγ (p = 0.131, 0.101, 0.002), NOS2 (p = 0.299, 0.142, 0.152), where TNFα maintained low expression levels across all treatments p = 0.299, 0.142, 0.152). Further, at 48 hours, IL-6 (p = 0.299, 0.142, 0.152), IFNγ (p = 0.071, < .001, 0.025), and TNFα (p = 0.019, < .001, < .001) all showed significant stepwise expression across increasing concentrations of nicotine. However, NOS2 showed an increase of expression at 25ng/ml, then a decrease at 50ng/ml, with a slight increase with 100ng/ml p = 0.044, 0.002, 0.013). At 72 hours, IL-6 (p = 0.052, 0.014, 0.001), IFNγ (p = 0.011, 0.004, 0.005), and TNFα (p = 0.004, 0.003, 0.003) showed a slight increase in expression at 100ng/ml. NOS2 (p = 0.003, 0.008, 0.003) showed a slight increase of expression with only 50ng/ml (**Fig 3**).

   At 24 hours, slight stepwise increases in expression were seen with IL-6 (p = 0.264, 0.278, 0.003), IFNγ (p = 0.131, 0.101, 0.002), NOS2 (p = 0.299, 0.142, 0.152), where TNFα maintained low expression levels across all treatments

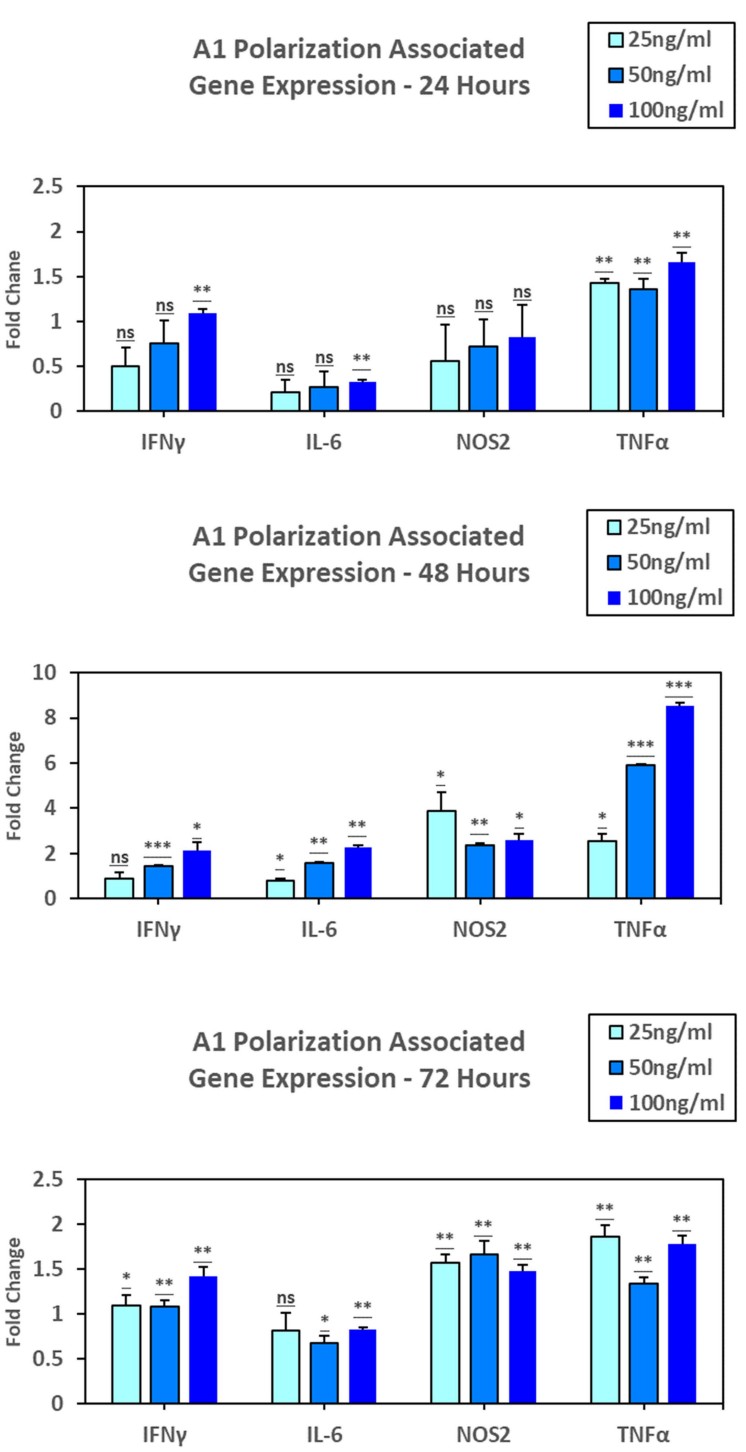

**Fig 3. Quantification of A1 polarization.**

p = 0.299, 0.142, 0.152). At 48 hours, IL-6 (p = 0.299, 0.142, 0.152), IFNγ (p = 0.071, <.001, 0.025), and TNFα (p = 0.019, <.001, <.001) all showed significant stepwise expression across increasing concentrations of nicotine. However, NOS2 showed an increase of expression at 25ng/ml, then a decrease at 50ng/ml, with a slight increase with 100ng/ml p = 0.044, 0.002, 0.013). At 72 hours, IL-6 (p = 0.052, 0.014, 0.001), IFNγ (p = 0.011, 0.004, 0.005), and TNFα (p = 0.004, 0.003, 0.003) showed a slight increase in expression at 100ng/ml. NOS2 (p = 0.003, 0.008, 0.003) showed a slight increase of expression with only 50ng/ml. These data suggest nicotine can drive A1 polarization of astrocytes.

**Nicotine exposure's effect on A2 polarization**

To determine the effect of nicotine on Astrocyte polarization and inflammation, C8D1A cells were treated with nicotine in vitro with 25ng/ml, 50ng/ml, and 100ng/ml nicotine for 24, 48, and 72 hours. Assessment of A2 markers at 24 hours indicated ARG expression increased with 50ng/ml exposure (p = 0.027, 0.002, 0.078), IL-10 expression increased with 100ng/ml exposure (p = 0.185, 0.023, 0.009), TGFβ showed a stepwise increase in expression from 25ng/ml to 100ng/ml (p = 0.016, 0.009, 0.014), and VEGF showed slight stepwise increase in expression from 25ng/ml to 100ng/ml (p = 0.087, 0.069, 0.111). At 48 hours, ARG expression appeared to double at 100ng/ml when compared to 25ng/ml (p = 0.01, 0.014, 0.002). IL-10 expression increased significantly at 100ng/ml exposure (p = 0.002, 0.005, <.001), TGFβ decreased with 50ng/ml and showed a slight increase in expression with 100ng/ml exposure (p = 0.035, 0.049, 0.036), and VEGF showed a similar decrease in expression from 25ng/ml to 50ng/ml with a slight increase with 100ng/ml (p = 0.576, 0.035, 0.153). At 72 hours, all cytokines showed a slight increase at 50ng/ml, IL-10 (p = 0.024, 0.011, 0.011) and TGFβ (p = 0.013, 0.065, 0.006) decreasing in expression at 100ngml and ARG (p = 0.018, 0.081, 0.006) and VEGF (p = 0.129, 0.011, 0.004) maintaining expression levels from 50ng/ml ([Fig 4]).

At 24 hours, ARG expression increased with 50ng/ml exposure (p = 0.027, 0.002, 0.078), IL-10 expression increased with 100ng/ml exposure (p = 0.185, 0.023, 0.009), TGFβ showed a stepwise increase in expression from 25ng/ml to 100ng/ml (p = 0.016, 0.009, 0.014), and VEGF showed slight stepwise increase in expression from 25ng/ml to 100ng/ml (p = 0.087, 0.069, 0.111). At 48 hours, ARG expression appeared to double at 100ng/ml when compared to 25ng/ml (p = 0.01, 0.014, 0.002). IL-10 expression increased significantly at 100ng/ml exposure (p = 0.002, 0.005, <.001), TGFβ decreased with 50ng/ml and showed a slight increase in expression with 100ng/ml exposure (p = 0.035, 0.049, 0.036), and VEGF showed a similar decrease in expression from 25ng/ml to 50ng/ml with a slight increase with 100ng/ml (p = 0.576, 0.035, 0.153). At 72 hours, all cytokines showed a slight increase at 50ng/ml, IL-10 (p = 0.024, 0.011, 0.011) and TGFβ (p = 0.013, 0.065, 0.006) decreasing in expression at 100ngml and ARG (p = 0.018, 0.081, 0.006) and VEGF (p = 0.129, 0.011, 0.004) maintaining expression levels from 50ng/ml to 100ng/ml. These data suggest nicotine that although not as consistent as A1 markers, nicotine can drive some A2 polarization of astrocytes.

## Discussion

Inclusive data establishes that astrocytes are sensitive to nicotine exposure, with nicotine eliciting a stimulatory effect on the cells at the 24- and 48-hour time points. Across all nicotine concentrations, this effect manifested as increased cell viability, which is interpreted as likely evidence of cellular proliferation.

Further analysis reveals an initial increase in apoptosis across all nicotine concentrations, with a return to homeostatic levels (comparable to control) at 18 hours. By 24 hours, minimal apoptosis was observed under control conditions, suggesting the cells had stabilized in culture. However, apoptosis activity persisted in nicotine-treated samples, particularly at concentrations ranging from 10 to 100 ng/ml. By 48 hours, apoptosis was significantly elevated, especially at a concentration of 25 mg/ml, indicating dose-dependent effects. This observation aligns with expectations, as increased cell density due to proliferation may trigger programmed cell death to maintain balance within the culture system [33–35].

Further, we aimed to assess the hypothesis that nicotine exposure affects the normal functioning of astrocytes, as measured by changes in mRNA expression. Previous research suggests that in astrocytes, as well as in other cell types

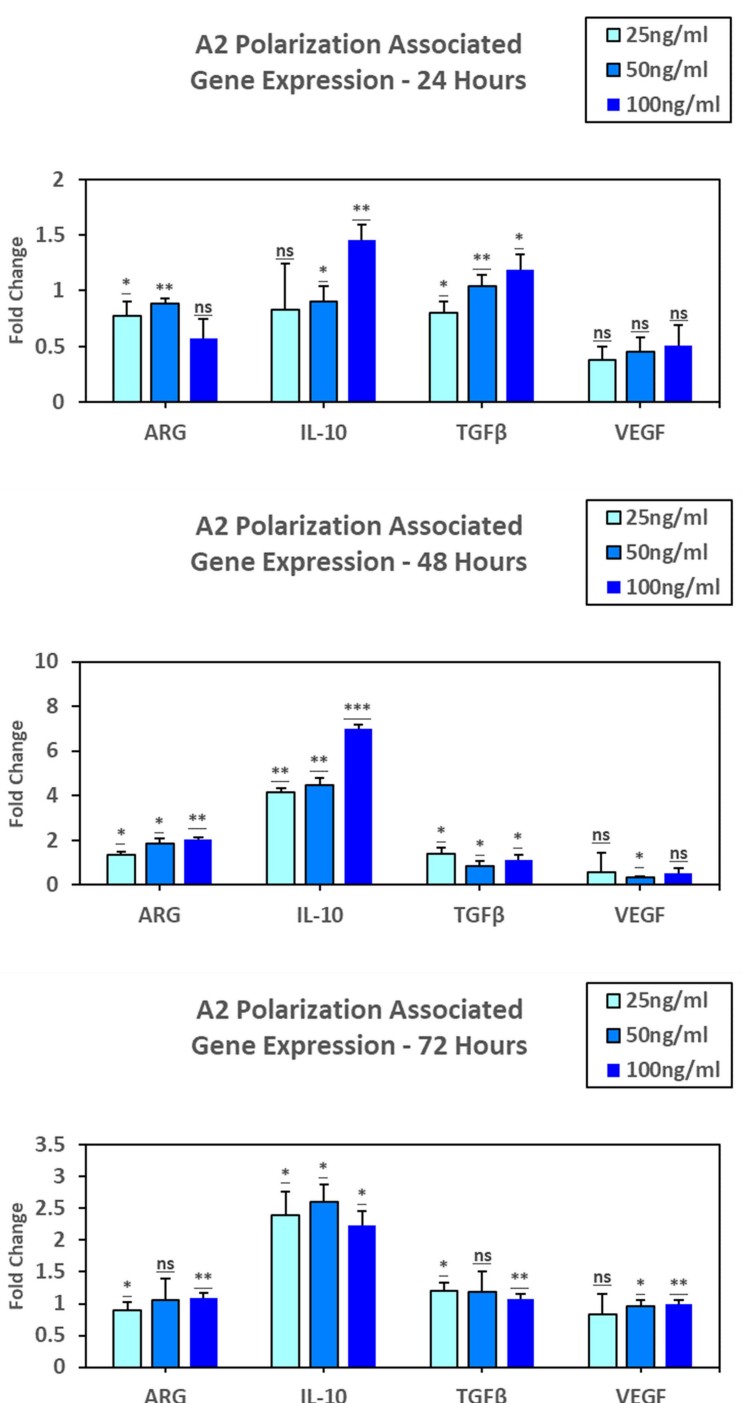

**Fig 4. Quantification of A2 polarization.**

like macrophages, markers often lack specificity for distinguishing between pro-inflammatory and pro-reparative activities [14,36,37]. Our findings are consistent with this, as nicotine exposure led to altered mRNA expression, but no clear distinction between A1 (pro-inflammatory) and A2 (anti-inflammatory) phenotypes was observed. This ambiguity may also be related to the selection of markers, although these markers (**Table 2**) are typically abundant during tissue insult and subsequent resolution to homeostasis [14,38–41].

For example, IL-6 is a well-established marker of inflammation but also possesses anti-inflammatory functions [10,42]. Specifically, IL-6 can inhibit TNFα production by macrophages when functioning in its anti-inflammatory role. Additionally, NOS2, which is induced by IFNγ, showed increased expression in our data. At the 24-hour time point, we observed elevated levels of both IL-6 and NOS2, with further increases at 50 ng/ml nicotine exposure at the 48- and 72-hour time points, suggesting induction of NOS2 by IFNγ.

Moreover, the observed decrease in ARG1 expression, a marker commonly associated with anti-inflammatory or pro-reparative responses [10,42] indicates a relative shift toward the A1 phenotype at 48 and 72 hours. This may be due to competition for nitric oxide (NO) between ARG1 and NOS2. Reduced ARG1 activity could reflect increased NO production by NOS2, as ARG1 normally functions to deplete NO from the cerebral environment.

Furthermore, ARG1 also plays an anti-inflammatory role by inhibiting TNFα, a pro-inflammatory cytokine that promotes glutamate release, which is critical for synaptic function and contributes to cerebral inflammation [4,9,10,43]. At 48 and 72 hours, ARG1 expression did not increase at the 50 ng/ml and 100 ng/ml nicotine concentrations, whereas TNFα expression exhibited a stepwise increase with higher nicotine concentrations. This suggests that nicotine may exacerbate pro-inflammatory signaling via TNFα, particularly under conditions of elevated exposure.

TGFβ, a pleiotropic growth factor [4,9,10,43] studied here, demonstrated decreased expression which correlates with our bioassays which showed decrease of cell proliferation. Stepwise increase in TGFβ expression at 24 hours 48 hour – increase at 100 ng/ml. 72 hour – increase at 50 ng/ml. This could be reflected in the stimulus influence of nicotine on the cells at the 24- and 48-hour time point with all concentrations of nicotine driving increases in proliferation, as seen in the functional assays.

Data herein suggests nicotine exposure may influence immune responses in the CNS, and that glia, specifically astrocytes, are responsive to nicotine exposure and react to this insult by morphological and signaling alterations. Previous research indicates a barrage of events follows the pro-inflammatory response (i.e., leukocyte infiltration, production of

**Table 2. Reference table for molecular targets studied.**

| Inflammatory Target | Type | Source | Function |
|---|---|---|---|
| IFNγ | Pro-inflammatory | Macrophages, T helper cells, Tc cells, B cells, NK cells | Promotes Th1 immune response – secretion of Th1 associated cytokines. |
| IL-6 | Anti-/Pro-inflammatory | B cells, T cells, monocytes | Pro- induces acute phase response and humoral immune response. Anti- inhibition of TNFα production by macrophages. |
| NOS2 | Pro-inflammatory; enzyme | Neurons, glia during inflammation | Induced by IFNγ; generates nitric oxide (NO) |
| TNFα | Pro-inflammatory | Macrophages, NK cells, B cells | Stimulates neutrophil activation, anticoagulant, tumor necrosis, stimulations adhesion molecules. |
| ARG | Anti-inflammatory enzyme | Macrophages | Removes excess NO; wound healing, tissue repair |
| IL-10 | Anti-inflammatory | Macrophages, monocytes, T cells, B cells | Inhibition of macrophage/monocyte and Th1cytokine production. |
| TGFβ-1 | Roles in both anti-/pro-inflammatory responses; growth factor | Microglia (strong polarizer) | Regulates many normal cell functions such as proliferation and cell death |
| VEGF | Potentially anti-/pro-inflammatory; Growth factor | Endothelial cells, macrophages, monocytes | Induces vascular permeability and macrophage activation |

pro-inflammatory cytokines) and is associated with activation of microglia and astrocytes that have the potential to contribute further to the inflammatory cascade [4,10,42,43]. Furthermore, astrocyte reactivity and subsequent responses may be dependent on the specific stimulus. Following nicotine exposure, the increase in pro-inflammatory cytokine production appears to coincide with an upregulation of pro-reparative processes. However, these reparative efforts do not appear sufficient to induce a full shift toward an A2 phenotype.

Astrocyte-targeted modulation of nicotine dependence-associated neural circuits presents a promising avenue for the development of novel smoking cessation strategies and offers broader insight into the neuropathological underpinnings of tobacco use disorder. Investigating astrocyte reactivity following chronic drug exposure may yield valuable therapeutic targets [46–49]. Nicotine use has been consistently associated with neuroinflammation, wherein reactive astrocytes serve as key mediators. Attenuating astrocyte-driven inflammatory responses may help ameliorate withdrawal symptoms and mitigate the neuroadaptive processes that underlie relapse.

One prominent pathological hallmark of nicotine dependence is the downregulation of the astrocytic glutamate transporter GLT-1 (EAAT2) within the nucleus accumbens, contributing to impaired glutamate clearance and excitotoxic signaling. Pharmacological agents such as ceftriaxone, a β-lactam antibiotic, and N-acetylcysteine (NAC), a cysteine prodrug with antioxidant and anti-inflammatory properties, have demonstrated efficacy in targeting astrocytic dysfunction. Ceftriaxone has been shown to upregulate GLT-1 expression, thereby restoring glutamate homeostasis and significantly reducing nicotine-seeking behaviors in preclinical models. Moreover, NAC modulates neuroimmune signaling pathways, including those in the nucleus accumbens, and has been observed to attenuate drug-seeking behavior. In addition to its role in glutamatergic regulation, ceftriaxone also decreases expression of pro-inflammatory markers such as tumor necrosis factor-alpha (TNF-α), potentially contributing to its anxiolytic effects during nicotine withdrawal. Collectively, these findings underscore the therapeutic potential of targeting astrocytic mechanisms in the treatment of nicotine addiction. Drugs that modulate astrocyte function, reduce reactivity, or promote astrocyte repair mechanisms might help mitigate the long-term effects of drug exposure on the brain [1,10,44–50].

Our ongoing research is progressing along multiple complementary lines, unified by the goal of elucidating astrocyte-specific responses to nicotine exposure. One key direction involves the application of next-generation sequencing to expand the repertoire of molecular markers, enabling a deeper understanding of acute nicotine-induced changes in astrocyte cellular signaling. This high-throughput approach aims to identify novel transcriptional and regulatory networks at both single-cell and population levels. Further nicotinic receptor subunit activity and modulators of these (including available pharmaceuticals such as varenicline and buproprion) should be specifically targeted to see if these represent robust targets of manipulations for therapeutic use., Concurrently, we are completing preliminary analyses on the effects of in utero nicotine exposure on glial cellularity and activity as the prenatal time period is a sensitive one for synaptogenesis and overall development of the brain. Using histological assessments during the early postnatal period, we are characterizing the enduring impact of prenatal nicotine exposure on astrocytes and microglia. These investigations seek to clarify how prenatal insults may alter glial development and function, potentially increasing susceptibility to neuropsychiatric conditions later in life.

Building upon these findings, future studies will extend to other glial populations, including oligodendrocytes, and examine intercellular interactions among glia and neurons. Particular emphasis will be placed on co-culture systems to explore the dynamic relationship between astrocytes and neurons under nicotine exposure and how this may affect neuroprotection and synaptic regulation.

Limitations include the limited markers studied for mRNA expression. Given the complexity of astrocyte phenotypes observed, which do not conform strictly to canonical A1 or A2 profiles, further resolution through flow cytometry or multi-omics array approaches is warranted. These methods will be instrumental in refining our understanding of astrocytic heterogeneity and state transitions following nicotine exposure, especially now that foundational expression data have been established.

Another limitation was simply the use of the murine in vitro model of study. However, in vivo studies represent a critical next phase, encompassing both developmental and adult models. Longitudinal investigations will be essential to determine how chronic nicotine exposure influences astrocyte reactivity over time and whether such exposure leads to persistent astrogliosis and the maintenance of a pro-inflammatory milieu. Ultimately, this work aims to delineate the cellular and molecular consequences of nicotine use, thereby informing the development of more effective therapeutic interventions to mitigate its public health burden.

## Conclusions

In conclusion, astrocytes, as highly adaptable glial cells, play a crucial role in sensing and influencing damaged neurons while integrating various signals to elicit specific responses that modulate neuroinflammation. The findings from this dissertation indicate that astrocytes exhibit different phenotypes at specific time points. Notably, based on the polarization data of A1 and A2 astrocyte phenotypes, there appears to be a threshold for nicotine concentration that induces a pro-inflammatory (A1) phenotype, particularly at a concentration of 100ng/ml. At lower concentrations, astrocyte reactivity demonstrates a diverse phenotype across different brain regions. The varied expression of inflammatory targets suggests that astrocytes may adopt either a neuroprotective or neurotoxic phenotype based on the specific requirements of the affected region, considering factors such as regional stimulation or inhibition of neighboring cells and the extent of insult. Further investigations are required to elucidate the activation of neurons and glial cells within the reward circuit. Although additional studies involving induced inflammation of astrocytes and the subsequent expression of A1 and/or A2 phenotypes were planned but not expanded upon in this dissertation, they hold promise for expanding our understanding of the continuum of astrocyte reactivity.

## Supporting information

**Supplementary Fig 1: Preliminary data used to establish an optimal cell density and determine the nicotine concentration levels of interest. Based on these cell viability assays, an optimal cell density of 2500 cells per well was determined.**
(DOCX)

**S1 File. Data**
(XLSX)

## Author contributions

**Conceptualization:** Leslie Sewell, James J Cray Jr.

**Data curation:** Leslie Sewell, James J Cray Jr.

**Formal analysis:** Leslie Sewell, James J Cray Jr.

**Funding acquisition:** James J Cray Jr.

**Investigation:** Leslie Sewell, James J Cray Jr.

**Methodology:** Leslie Sewell, James J Cray Jr.

**Project administration:** James J Cray Jr.

**Resources:** James J Cray Jr.

**Software:** James J Cray Jr.

**Supervision:** James J Cray Jr.

**Validation:** Leslie Sewell, James J Cray Jr.

**Visualization:** Leslie Sewell, James J Cray Jr.

**Writing – original draft:** Leslie Sewell, James J Cray Jr.

**Writing – review & editing:** Leslie Sewell, James J Cray Jr.

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
