## [Decision Letter · Decision Letter 0]

PONE-D-24-55873Nicotine alters cellular activity and mRNA expression of patterns in vitro of murine astrocytesPLOS ONE

Dear Dr. Cray Jr.,

Thank you for submitting your manuscript to PLOS ONE. After careful consideration, we feel that it has merit but does not fully meet PLOS ONE’s publication criteria as it currently stands. Therefore, we invite you to submit a revised version of the manuscript that addresses the points raised during the review process.

The conclusions need to be more precise and be in line with the here shown data. There are some issues regarding statistical significance. The introduction needs to be expanded, and the context to smoking and nicotine abuse needs to be clear. 

We look forward to receiving your revised manuscript.

Kind regards,

Henning Ulrich

Academic Editor

PLOS ONE

Journal Requirements:

The Ohio State University College of Medicine (JC)

3.Please include captions for your Supporting Information files at the end of your manuscript, and update any in-text citations to match accordingly. Please see our Supporting Information guidelines for more information: http://journals.plos.org/plosone/s/supporting-information.

Reviewers' comments:

Reviewer's Responses to Questions

**Comments to the Author**

1. Is the manuscript technically sound, and do the data support the conclusions?

Reviewer #1: Yes

Reviewer #2: Yes

Reviewer #3: Yes

2. Has the statistical analysis been performed appropriately and rigorously? 

Reviewer #1: Yes

Reviewer #2: Yes

Reviewer #3: I Don't Know

3. Have the authors made all data underlying the findings in their manuscript fully available?

Reviewer #1: Yes

Reviewer #2: Yes

Reviewer #3: Yes

4. Is the manuscript presented in an intelligible fashion and written in standard English?

Reviewer #1: Yes

Reviewer #2: Yes

Reviewer #3: Yes

5. Review Comments to the Author

Reviewer #1: Comments:

The manuscript addresses an important topic. The study tests the effect of nicotine on murine astrocytes, focusing on cellular activity and mRNA expression. The results indicate that Nicotine impacts astrocyte activity and inflammatory signaling, suggesting its potential involvement in neuroinflammation. For better clarity the following revision is recommended:

1. The abstract as well as introduction should be modified to describe Tobacco/Nicotine consumption as a major global health concern/risk in general (in adults and in pregnancy), as nicotine teratological effect in astrocytic embryonic development was not specifically assessed in this paper. C8D1A cell line was used (Postnatal d8 astrocytic cerebellum).

2. The introduction section should be expanded to include:

a. Nicotine absorption, concentration in blood and brain, half-life (2 hours) etc. This would help understand the relevancy of selected doses and culture feeding regimen used in the study.

b. Summary of research exploring the effect of Nicotine on astrocytes should be expanded, include studies demonstrating the expression of nicotinic acetylcholine receptors (nAChRs) in astrocytes, and its downstream effects on astrocytic signaling pathways, morphological and functional changes etc.

Some recommended refs:

a. Aryal et al, Glia. 2021 Apr 14;69(8):2037–2053. doi: 10.1002/glia.24011

b. Stellwagen et al 2019, Curr Opin Neurobiol 57: 179–185. doi.org/10.1016/j.conb.2019.02.010.

c. Hernández-Morales et al, 201, Neuroscience 2014

3. Material and Methods:

a. Please include media change regimen (e.g. daily? Every other day? Etc) – might affect results interpretation (Acute exposure vs. chronic).

b. Include Nicotine source and Cat number.

4. Results:

a. MTS assay: The assay relies on cellular metabolic activity, which may not always directly correlate with cell number, if feasible please correlate with number of counted cells (e.g. neuclocounter/Hemcytometer, Brdu, ICF Dapi count), if not applicable, address this issue in discussion section.

b. Figure 1. Where applicable please add statistical significance asterisks (e.g. in comparison with non-treated-NT arm (0ng/ml) at same time point). The Y axis should be modifies to represent Cell Viability (measured by OD).

c. The reduction in cell activity/cell number or metabolic activity in un-treated arm after 24 & 48 hours compared to 12&18hr should be explained.

d. Figure 2. – Where applicable statically significance (asterisks) compared to NT should be added, Y-axis (apoptosis – measured by RFU – add abbreviation (Relative Fluorescence Units).

e. The peak in mRNA expression (in both A1/A2 markers graphs) occurs after 48 hr (higher fold change – pls note difference in scaling), should be discussed/explained.

Discussion section:

1. The effect of Nicotine was assessed only in one murine cell line (derived from Cerebellum), The potential difference in Nicotine effect between human and mouse astrocytes, as well as astrocytes heterogenous sub-populations (region/functionality/morphological) should be discussed.

2. Discuss the extrapolation between study dosing and feeding regimen and potential in-vivo consumption

3. Propose directions/strategies to overcome the resolution of A1/A2 marker expression and its cross-talk with other CNS cell population, such as including additional markers, multi-omics, in-vivo analysis.

Reviewer #2: The manuscript addresses a significant public health concern by investigating the effects of nicotine on murine astrocytes, focusing on cellular activity and mRNA expression. The study is well-designed, employing appropriate methodologies to assess astrocyte responses to nicotine exposure. However, certain areas require further refinement to enhance clarity, rigor, and relevance to the broader scientific and public health community.

Specific Comments on Sections

1. Introduction

o The introduction discusses the effects of nicotine during pregnancy, but it is unclear how this background connects to the study’s objectives. The authors should clarify:

Why is the focus on nicotine exposure during pregnancy relevant to astrocyte activity in this experimental model?

How does the current study design address the broader implications for pregnancy-related outcomes, if at all?

If the focus is more general (e.g., nicotine's effects on the central nervous system), consider rephrasing or narrowing the discussion for better alignment with the study's aim.

o May need reference for “Discuss how findings from murine cells can be cautiously extrapolated to human physiology and potential limitations of this approach”.

2. Materials and Methods

o Cell Model:

Why were murine astrocytic cells chosen instead of human astrocytic cells?

Discuss how findings from murine cells can be cautiously extrapolated to human physiology and potential limitations of this approach.

o Dosage Selection:

Clarify the rationale for the chosen nicotine dosages at each experimental step.

Provide references or scientific justification for selecting the initial concentrations (0, 10, 25, 50, 100, 250, 500 ng/mL) and explain why the later experiments only used 0, 25, 50, and 100 ng/mL.

How do these concentrations relate to physiologically relevant exposures in humans (e.g., plasma nicotine levels in smokers or vapers)?

o Experimental Details:

Ensure that dosages and time points are explicitly mentioned for each assay. This will help readers replicate and interpret the study.

3. Results

o The description of findings is generally clear, but some terminology, such as "diminishing returns were observed," requires more explicit explanation for readers who may not be familiar with the concept.

o Provide a brief interpretation of how observed changes in apoptosis and proliferation reflect potential mechanisms of nicotine's impact on astrocytes.

4. Discussion

o Expand on the potential therapeutic implications of the findings (if possible):

How might astrocyte modulation contribute to strategies for addressing nicotine addiction and neuroinflammatory disorders?

Could targeting astrocytes play a role in smoking cessation interventions or reduce the neurological impact of nicotine exposure?

o Broaden the discussion to include potential applications of the study's findings beyond pregnancy, such as the relevance of astrocyte responses to nicotine in adolescent brain development or chronic nicotine users if possible.

Reviewer #3: In this work, the author used different functional assays to test nicotine effects on C8D1A cell viability, proliferation, and apoptosis at different time points. They also demonstrated the nicotine effects on gene expression of pro/anti-inflammatory markers. However, there lacks evidence to support the conclusion that nicotine alters astrocyte activity and inflammatory signaling. In the meanwhile, the statistical analysis in this research is not clear. The concerning are listed below:

Major

• Figure 1, your result suggests that excessive nicotine may lead to diminishing effects (no additional effects) based on the MTS assay. Have your considered testing higher dose to further explore this trend? Usually, 0.1-10µM (16-1620 ng/mL) dose of nicotine mimic as regular smoking or high environmental exposure.

• Figure 2, what is the reason for high apoptosis level in control group (0 ng/mL of nicotine) at 12, 18, and 48 hours? And there is no apoptosis measured in the same condition at 24 hours.

• In figure 2 legend, "Note similar relationship for nicotine concentration to that observed for MTS assay," please clarify or provide more information for this conclusion. In figure 1 MTS assay, the cell viability increased by nicotine treatment at lower dose at the 48-hour treatment group. However, the figure 2 shows that the apoptosis level significantly increased by nicotine treatment at lower dose at the 48-hour treatment group.

• In your caspase assay, there is a significant increase in apoptosis with the 10 ng/mL nicotine treatment at 12 hours. To strengthen your findings, consider including gene expression data corresponding to the 10 ng/mL treatment at 12 hours for Figures 3 and 4.

• For figure 3 and 4, clarity whether the significant differences are between different groups (e.g., dose comparisons) or between the control group and each treatment group.

Minor:

• Please specify the statistical methods used for analysis in the figure legends and in the research method. And specify whether the error bars represent standard error of the mean or standard deviation.

• For figure 1 and 2, include significance markers (e.g., stars) to indicate statistically significant changes.

• Define the p-value thresholds for the significance levels.

• Figure 3 and 4, please clarify the experiment number and independent repeat number for the representative blots as shown in the figures.

6. PLOS authors have the option to publish the peer review history of their article (what does this mean? ). If published, this will include your full peer review and any attached files.

**Do you want your identity to be public for this peer review?** For information about this choice, including consent withdrawal, please see our Privacy Policy .

Reviewer #1: No

Reviewer #2: **Yes: ** Li Feng

Reviewer #3: No

---

## [Author Response · Author response to Decision Letter 1]

23 Apr 2025

Response to Reviewers:

GENERAL COMMENTS

Reviewer I:

• The manuscript addresses a significant public health concern by investigating the effects of nicotine on murine astrocytes, focusing on cellular activity and mRNA expression. The study is well-designed, employing appropriate methodologies to assess astrocyte responses to nicotine exposure. However, certain areas require further refinement to enhance clarity, rigor, and relevance to the broader scientific and public health community.

Reviewer II:

• The manuscript addresses an important topic. The study tests the effect of nicotine on murine astrocytes, focusing on cellular activity and mRNA expression. The results indicate that Nicotine impacts astrocyte activity and inflammatory signaling, suggesting its potential involvement in neuroinflammation.

Specific Comments on Sections

ABSTRACT

Reviewer II:

• The abstract as well as introduction should be modified to describe Tobacco/Nicotine consumption as a major global health concern/risk in general (in adults and in pregnancy), as nicotine teratological effect in astrocytic embryonic development was not specifically assessed in this paper. C8D1A cell line was used (Postnatal d8 astrocytic cerebellum).

Response: We have updated the abstract and introduction to reflect this request. We note the cells were obtained as a cell line and maintained from a postnatal model. Most of our prior work on nicotine has been in the prenatal developmental sphere and we apologize for not taking a closer editing eye.

INTRODUCTION

Reviewer I:

• The introduction discusses the effects of nicotine during pregnancy, but it is unclear how this background connects to the study’s objectives. The authors should clarify:

• Why is the focus on nicotine exposure during pregnancy relevant to astrocyte activity in this experimental model? Response: As above it is not. We are interested in astrocyte reaction to any exposure including prenatal and have updated throughout to reflect this as the reviewer identified our approach was neither specific nor limited to a pregnancy exposure model.

• How does the current study design address the broader implications for pregnancy-related outcomes, if at all? Response: We have attempted to address this now in the discussion as we do have ongoing experiments assessing prenatal exposures on glial cells in vivo via a histological approach. As above, we apologize we were not more careful in our first submission.

• If the focus is more general (e.g., nicotine's effects on the central nervous system), consider rephrasing or narrowing the discussion for better alignment with the study's aim. Response: We hope our edits reflect our appreciation for the reviewer bringing this to our attention.

• May need reference for “Discuss how findings from murine cells can be cautiously extrapolated to human physiology and potential limitations of this approach”. Response: That is a fantastic point and not simply limited to murine to human comparison but also cell line vs primary cell approaches. We wanted a robust high throughput system for study and chose the murine astrocyte as it is an established cell line and with the fore knowledge, we were conducting prenatal exposure in an in vivo model we planned to study histologically for glial cell effects. We have added this to discussion, limitations, and future directions.

Reviewer II:

• The introduction section should be expanded to include:

• Nicotine absorption, concentration in blood and brain, half-life (2 hours) etc. This would help understand the relevancy of selected doses and culture feeding regimen used in the study. Response: Thank you for this comment. The selected feeding program was based on previous work by our own group on mesenchymal cell sensitivity to nicotine exposure, correlative cotinine concentrations for active nicotine use (30-300ng/ml) and our inclusive bioassays (FIGURE 1 and 2). We utilized both MTS and Apoptosis assay to define dose concentration and timepoints to be used for mRNA studies. Our range after considering IC50 and EC50 values led us to the 25-100ng/ml dose range that was used.

* Tobacco smoke comprises a complex mixture of hundreds of chemical compounds, many of which may potentiate the psychoactive properties of nicotine. Nonetheless, nicotine remains the principal agent underlying tobacco dependence, primarily through its capacity to reinforce drug-seeking and drug-taking behaviors. A key factor contributing to this dependence is nicotine’s relatively short plasma half-life of approximately two hours, indicating that half of the administered dose is metabolized and cleared from the body within that timeframe [10]. This rapid pharmacokinetic profile results in a transient duration of action, often compelling individuals to engage in frequent re-administration to sustain its psychoactive effects.

• Summary of research exploring the effect of Nicotine on astrocytes should be expanded, include studies demonstrating the expression of nicotinic acetylcholine receptors (nAChRs) in astrocytes, and its downstream effects on astrocytic signaling pathways, morphological and functional changes etc. Response: Thank you. We had previously edited this down for brevity but have included this now in discussion. We are relying primarily on published literature as in our own laboratory we have only conducted experiments with the alpha7 subunit for obvious reasons as it homodimerizes and is involved in addiction. We have expanded our discussion accordingly.

MATERIALS AND METHODS

Reviewer I:

• Cell Model:

• Why were murine astrocytic cells chosen instead of human astrocytic cells? Response: We are a preclinical teratology focused. Our next step is to use a translational in utero exposure model, so our preliminary experimentations were done with murine astrocyte cells.

• Discuss how findings from murine cells can be cautiously extrapolated to human physiology and potential limitations of this approach. Response: We have updated the discussion to address these concerns.

* Rationale for Using Murine Astrocytic Cells:

* Murine astrocytic cells (C8D1A) were selected for this study due to their well-characterized, reproducible behavior in vitro and their widespread use as a model for studying astrocyte physiology and response to pharmacological agents. These cells provide a controlled and genetically stable platform for investigating mechanisms of cellular response, including proliferation, apoptosis, and gene expression. While human astrocytes offer species-specific insights, they present limitations such as donor variability, ethical constraints, and reduced scalability for high-throughput assays. Moreover, murine models remain highly relevant for translational research, as they are commonly used in in vivo studies of neuroinflammation, addiction, and neurodegeneration, facilitating continuity between in vitro and in vivo experimentation.

• Dosage Selection (ADDRESSED IN INTRO):

• Clarify the rationale for the chosen nicotine dosages at each experimental step.

* Given the limited data on the specific effects of nicotine on astrocytes, we conducted targeted in vitro studies to determine whether astrocytes are responsive to nicotine and whether nicotine exposure alters their cell cycle in a dose- and time-dependent manner. Based on reported plasma nicotine concentrations following the consumption of a single cigarette (5–30 ng/mL; Benowitz & Jacob, 1994), we selected a range of nicotine concentrations (0, 10, 25, 50, 100, 250, and 500 ng/mL) to capture a spectrum of cellular responses. Subsequent analyses focused on concentrations (0, 25, 50, and 100 ng/mL) that elicited measurable cellular activity. In parallel, we assessed astrocytic mRNA expression profiles to test the hypothesis that nicotine exposure shifts the balance between neurotoxic/pro-inflammatory and neuroprotective/pro-reparative markers.

• Provide references or scientific justification for selecting the initial concentrations (0, 10, 25, 50, 100, 250, 500 ng/mL) and explain why the later experiments only used 0, 25, 50, and 100 ng/mL. Response: These doses allowed for maximal effect without unnecessary toxicity.

* After preliminary experimentation, two concentrations of nicotine, namely 25ng/ml and 50ng/ml, were selected based on their significance as determined by cell viability assays. Cell viability, indicative of proliferation, was evaluated using the colorimetric MTS assay. Apoptosis was assessed using the Apo-ONE Caspase-3/7 fluorescence assay. The results from cell viability assays led to the determination of 25ng/ml and 50ng/ml being those concentrations of interest. The 100ng/ml nicotine concentration was added to gauge effects of an exorbitant exposure of nicotine. We also decided to explore these concentrations at 24-, 48-, and 72-hour time points.

• How do these concentrations relate to physiologically relevant exposures in humans (e.g., plasma nicotine levels in smokers or vapers)? RESPONSE: Literature suggests 30-300 ng/ml is indicative of active nicotine use.

* Active smokers typically consume between 10 and 100 mg of nicotine per day, while individuals using alternative nicotine delivery systems may be exposed to an even broader range of nicotine levels (Benowitz 1984; L Benowitz, et al 1988)

• Experimental Details:

• Ensure that dosages and time points are explicitly mentioned for each assay. This will help readers replicate and interpret the study. Response: This has been updated throughout.

Reviewer II:

• Material and Methods:

• Please include media change regimen (e.g. daily? Every other day? Etc.) – might affect results interpretation (Acute exposure vs. chronic). All regimens were acute and only necessitated one feed as indicated by the initial dose schema gating of the MTS and Caspase assays (Figures 1 and 2)

• Include Nicotine source and Cat number.

* Sigma Life Science (−)-Nicotine ≥99% (GC), liquid N3876-100ml CAS Number 54-11-5 (IN TEXT: Sigma Aldrich, St. Louis, MO, USA, N3876)

RESULTS

Reviewer I:

• The description of findings is generally clear, but some terminology, such as "diminishing returns were observed," requires more explicit explanation for readers who may not be familiar with the concept. Response: We have updated our language for consistency.

* C8D1A cell viability following nicotine exposure was assessed using the MTS assay at 12, 18, 24, and 48 hours. Early time points (12 and 18 hours) revealed a modest reduction in cell viability across most nicotine concentrations, with a slight rebound observed at 100 ng/mL at 12 hours and a similar trend at 18 hours following 50 ng/mL treatment. These findings suggest an initial sensitivity of astrocytes to nicotine. By 24 and 48 hours, nicotine exposure exhibited a stimulatory effect on cell proliferation across all tested concentrations. However, this proliferative response plateaued at concentrations above 50 ng/mL, indicating that increasing nicotine concentration beyond 50 ng/mL did not result in proportionally greater increases in cell proliferation—the beneficial effect of nicotine started to level off, or even decrease, despite higher doses (Figure 1)

• Provide a brief interpretation of how observed changes in apoptosis and proliferation reflect potential mechanisms of nicotine's impact on astrocytes. Response: Although we did interpret the time course in this vein the initial experiments were simply conducted as a pharmacological bioassay to define critical concentrations to be tested for mRNA marker studies focused on astrocyte polarization.

Reviewer II:

• MTS Assay:

• The assay relies on cellular metabolic activity, which may not always directly correlate with cell number, if feasible, please correlate with number of counted cells (e.g., nucleocounter/Hemocytometer, ICF, DAPI Count). If not applicable, address this issue in the discussion section. Response: We have added this to the limitations, and although we agree MTS is still a standard in the field of experimental pharmacology and drug development to define IC25/50/75/90 and EC25/50/75/90 values, etc. We used these tools is a similar fashion to identify critical concentration schema for polarization studies.

* MTS assays were utilized in our cell culture studies to obtain a more accurate and functionally relevant assessment of cell viability. Unlike direct cell counting methods, which merely quantify total cell numbers, the MTS assay provides insight into cellular metabolic activity. Aside from determining initial seeding densities, cell counts were not employed, as they do not adequately reflect the metabolic status or viability of the cultured cells.

• Figure 1. Where applicable please add statistical significance asterisks (e.g. in comparison with non-treated-NT arm (0ng/ml) at same time point). The Y axis should be modified to represent Cell Viability (measured by OD). Response: We would be happy to revisit this if considered necessary given our above explanation of how we utilized our MTS and apoptosis information.

• The reduction in cell activity/cell number or metabolic activity in un-treated arm after 24 & 48 hours compared to 12&18hr should be explained. Response: This is likely an artifact of cell culture on plastic, cell loss is often observed in the first 24-48 hours after plating which is why control 0ng/ml group is so critical for interpretation.

• Figure 2. Where applicable statical significance (asterisks) compared to NT should be added, Y-axis (apoptosis – measured by RFU – add abbreviation (Relative Fluorescence Units). Response: We would be happy to revisit this if considered necessary given our above explanation of how we utilized our MTS and apoptosis information.

• Figure 3 & 4: The peak in mRNA expression (in both A1/A2 markers graphs) occurs after 48 hr. (higher fold change – pls note difference in scaling), should be discussed/explained. Response: As we are not comparing between timepoints we did not use the same y axis scale. We would be happy to revisit. However, we interpret the highest values changes as being the critical signaling axis peak after acute exposure. We have added this to the discussion.

DISCUSSION

Reviewer I:

• Expand on the potential therapeutic implications of the findings (if possible):

• How might astrocyte modulation contribute to strategies for addressing nicotine addiction and neuroinflammatory disorders? Response: We would like to thank the reviewer for prompting our expansion of the discussion. Astrocytes have been linked to drug seeking behaviors and transcriptional changes are crucial to interpreting biological responses to nicotine we do believe that the astrocyte may prove a useful target in the monitoring of inflammatory mechanistic processes related to spectrum of nicotine related health problems.

• Could targeting astrocytes play a role in smoking cessation interventions or reduce the neurological impact of nicotine exposure? Response: As astrocytes play a critical role in nicotine use and cessation relapse, we do believe that targeted therapies that alter extracellular levels of certain neurotransmitters (i.e. dopamine) associated with astrocyte activity may prove to be a useful therapeutic tool.

• Broaden the discussion to include potential applications of the study's findings beyond pregnancy, such as the relevance of astrocyte responses to nicotine in adolescent brain development or chronic nicotine users if possible. Response: We agree our focus was previously too heavily reliant on interpretations of prenatal exposures. We have updated throughout.

Reviewer II:

• The effect of Nicotine was assessed only in one murine cell line (derived from Cerebellum), The potential difference in Nicotine effect between human and mouse astrocytes, as well as astrocytes heterogenous sub-populations (region/functionality/morphological) should be discussed. Response: Agreed. This is similar to a previously addressed comment above.

• Discuss the extrapolation between study dosing and feeding regimen on potential in-vivo consumption. Response: Agreed. This is similar to a previously addressed comment above.

• Propose directions/strategies to overcome the no specificity of A1/A2 markers, such as including additional markers, multi-omics, in-vivo analysis. Respon

---

## [Decision Letter · Decision Letter 1]

Nicotine alters cellular activity and mRNA expression of patterns of astrocytes

PONE-D-24-55873R1

Dear Dr. Cray Jr.,

We’re pleased to inform you that your manuscript has been judged scientifically suitable for publication and will be formally accepted for publication once it meets all outstanding technical requirements.

Kind regards,

Henning Ulrich

Academic Editor

PLOS ONE

Additional Editor Comments (optional):

Reviewers' comments:

Reviewer's Responses to Questions

**Comments to the Author**

1. If the authors have adequately addressed your comments raised in a previous round of review and you feel that this manuscript is now acceptable for publication, you may indicate that here to bypass the “Comments to the Author” section, enter your conflict of interest statement in the “Confidential to Editor” section, and submit your "Accept" recommendation.

Reviewer #1: All comments have been addressed

Reviewer #2: All comments have been addressed

2. Is the manuscript technically sound, and do the data support the conclusions?

Reviewer #1: Yes

Reviewer #2: Yes

3. Has the statistical analysis been performed appropriately and rigorously? 

Reviewer #1: Yes

Reviewer #2: Yes

4. Have the authors made all data underlying the findings in their manuscript fully available?

Reviewer #1: Yes

Reviewer #2: Yes

5. Is the manuscript presented in an intelligible fashion and written in standard English?

Reviewer #1: Yes

Reviewer #2: Yes

6. Review Comments to the Author

Reviewer #1: Dear Authors,

Thank you for your thoughtful revisions. You've addressed the comments well, and the manuscript has improved significantly. I’m pleased to see it moving forward toward publication.

Wishing you continued success with your work.

Reviewer #2: Thank you for your thorough revisions. I have re-reviewed the manuscript and confirm that all previously raised concerns have been addressed appropriately. The manuscript is clear, scientifically sound, and ready for publication.

7. PLOS authors have the option to publish the peer review history of their article (what does this mean? ). If published, this will include your full peer review and any attached files.

**Do you want your identity to be public for this peer review?** For information about this choice, including consent withdrawal, please see our Privacy Policy .

Reviewer #1: **Yes: ** Michal Izrael

Reviewer #2: **Yes: ** Li Feng

---

## [Editor Report · Acceptance letter]

PONE-D-24-55873R1

PLOS ONE

Dear Dr. Cray Jr.,

I'm pleased to inform you that your manuscript has been deemed suitable for publication in PLOS ONE. Congratulations! Your manuscript is now being handed over to our production team.

Kind regards,

on behalf of

Dr. Henning Ulrich

Academic Editor

PLOS ONE